# Foxp3+ regulatory T cells maintain the bone marrow microenvironment for B cell lymphopoiesis

Antonio Pierini[1,2,*], Hidekazu Nishikii[1,3,*], Jeanette Baker[1], Takaharu Kimura[4], Hye-Sook Kwon[1], Yuqiong Pan[1], Yan Chen[1], Maite Alvarez[1], William Strober[1], Andrea Velardi[2], Judith A. Shizuru[1], Joy Y. Wu[4], Shigeru Chiba[3] & Robert S. Negrin[1]

Foxp3+ regulatory T cells (Treg cells) modulate the immune system and maintain self-tolerance, but whether they affect haematopoiesis or haematopoietic stem cell (HSC)-mediated reconstitution after transplantation is unclear. Here we show that B-cell lymphopoiesis is impaired in Treg-depleted mice, yet this reduced B-cell lymphopoiesis is rescued by adoptive transfer of affected HSCs or bone marrow cells into Treg-competent recipients. B-cell reconstitution is abrogated in both syngeneic and allogeneic transplantation using Treg-depleted mice as recipients. Treg cells can control physiological IL-7 production that is indispensable for normal B-cell lymphopoiesis and is mainly sustained by a subpopulation of ICAM1+ perivascular stromal cells. Our study demonstrates that Treg cells are important for B-cell differentiation from HSCs by maintaining immunological homoeostasis in the bone marrow microenvironment, both in physiological conditions and after bone marrow transplantation.

[1] Department of Medicine, Division of Blood and Marrow Transplantation, Stanford University, 269W. Campus Drive, Stanford, California 94305, USA. [2] Department of Medicine, Hematopoietic Stem Cell Transplantation Program, University of Perugia, Piazzale Menghini, 06132, Sant'Andrea delle Fratte, Perugia 06132, Italy. [3] Department of Hematology, Faculty of Medicine, University of Tsukuba, 1-1-1 Tennodai, Tsukuba, Ibaraki 305-8575, Japan. [4] Department of Medicine, Division of Endocrinology, Stanford University of Medicine, 300 Pasteur Drive, Stanford, California 94305, USA. * These authors contributed equally to this work. Correspondence and requests for materials should be addressed to R.S.N. (email: negrs@stanford.edu).

Foxp3[+] regulatory T cells (Treg cells) have an important role in immune homeostasis and provide protection from autoimmune diseases. Foxp3 is specifically expressed in CD4[+]CD25[+] Treg cells and is required for the differentiation from naive CD4[+] T cells[1]. Loss-of-function mutations of *foxp3* gene in both humans and mice results in the lack of Treg cells and development of autoimmune diseases[1,2]. Treg cells are also important for modulating complications of allogeneic transplantation, such as Graft-versus-host-disease (GVHD). Co-infusion of freshly-isolated donor Treg cells can abrogate GVHD without reducing graft-versus-tumor (GVT) effects both in mouse models and in humans[3,4]. On the other hand, the role of host-derived Treg cells after transplantation is unclear[5]. Utilizing *in vivo* imaging, host-derived Treg cells co-localizes with infused allogeneic haematopoietic stem cells (HSC), suggesting a possible role for Treg cells in providing an immune niche to HSCs helping them evade host immunity, and favouring their survival[6].

Within the HSC-niche, cellular components maintain and regulate HSC 'stemness'. The HSC-niche is thought to be a perivascular area in the bone marrow (BM), created by mature haematopoietic cells, mesenchymal stem cells (MSC), stromal cells, endothelial cells, osteoblasts/osteoclasts, sympatic nerves and non-myelinating Schwann cells. Niche dysfunction in any of these components might induce HSC loss or functional defects. Perivascular stromal cells, a component of the HSC-niche, secrete CXCL12 and other growth factors important for HSC homing and further differentiation into specific cell lineages[7,8].

In GVHD experimental models, it has been reported that alloreactive Fas[+] T cells can target some components of the HSC-niche after transplantation to induce a defect in lymphoid differentiation from HSC mainly affecting B cells[9]. Here, we show that the *in vivo* depletion of host-derived Treg cells induces expansion of the phenotypic long-term HSC population (CD34[−] Lineage[−] cKit[+] Sca1[+] population), thereby reducing the production of B cell progenitors and mature B cells. Moreover, severe defects of donor-derived B lymphopoiesis are detected after syngeneic/allogeneic transplantation in Treg-depleted recipients. Lastly, we find that the perivascular ICAM1[+]CD31[−]CD45[−]TER119[−] stromal cells located in the BM have reduced Interleukin-7 (IL-7) and CXCL12 production after Treg depletion suggesting that activated T cells that are generated in the absence of Treg cells may target lineage-specific BM niche cells, resulting in defective lymphopoiesis from HSC. Therefore, our results suggest that Treg cells regulate the production of important growth factors for lymphopoiesis and are crucial for maintaining niche activity and for preserving the function of HSC. These results provide new insights into Treg cells biology and function and are also relevant for further clinical application for the modulation of immune reconstitution defects after transplantation.

## Results

**Treg depletion induces a B-cell differentiation defect**. To evaluate the impact of Foxp3[+] Treg cells on haematopoiesis in the BM, we analysed BM stem and progenitor cells in Foxp3-DTR (FTR) mice with or without treatment with diphtheria toxin (DT). FTR mice received 1 μg DT every other day for five injections (Fig. 1a). We next analysed the phenotype and the number of several immune cells and progenitor populations, including myeloid cells (Gr1[+], Mac1[+]), B cells and B-cell progenitors. We found an increase in the Gr1[+]Mac1[+] cells ($P < 0.01$, Student's *t*-test) and a significant decrease in the frequency of B220[+] B cells ($P < 0.001$, Student's *t*-test) in the BM of Treg-depleted mice (Fig. 1b–d). The total number or

BM cells after DT treatment were slightly increased in Treg-depleted mice (Fig. 1e). Mature B cells (IgM[+]B220[+], $P < 0.001$, Student's *t*-test), Pre-B cells (IgM[−]B220[+]CD19[+] cKit[−], $P < 0.001$, Student's *t*-test) and Pro-B cells (IgM[−]B220[+] CD19[+]cKit[+], $P < 0.05$, Student's *t*-test) were all decreased with a slight increase in PrePro-B cells (IgM[−]B220[+]CD19[−]cKit[−] Flt3[+], $P < 0.05$, Student's *t*-test, Fig. 1f–h).

In the haematopoietic stem/progenitor cell fraction, we detected a significant increase in the frequencies of haematopoietic stem/progenitor cells (Lin[−]Sca1[+]cKit[+] cells; LSK, $P < 0.001$, Student's *t*-test, Fig. 1i,j), lymphoid primed multipotent progenitor (Flt3[high +]LSK; LMPP, $P < 0.0001$, Student's *t*-test, Fig. 1k), and long-term HSC population (CD34[−] LSK, $P < 0.001$, Student's *t*-test, Fig. 1l) in mice depleted of Treg cells with DT. Moreover, analysing cell cycle status, the $G_0$ state of LSK and CD34[−]LSK (Ki67[−] Hoechst33324[−] population) was significantly reduced in Treg-depleted FTR mice (Fig. 1m–o). There were no significant differences in WT mice with or without DT treatment, suggesting that these results were not due to non-specific toxicity of DT treatment (Fig. 1). According to these data, we conclude that Treg depletion induces a block in the early phase of the B cell differentiation process, while phenotypic HSC populations enter into cell cycle from the quiescent state and expand.

We also analysed BM T cells in the presence and absence of Treg cells. Although the total number of T cells was not significantly different, the frequencies of CD62L[+]CD44[−] naive T cells in the CD4[+] and CD8[+] T-cell subpopulations were decreased ($P < 0.01$, Student's *t*-test), while CD44[+]CD62L[−] effector T cells were increased in Treg-depleted mice ($P < 0.01$, Student's *t*-test, Supplementary Fig. 1A–C). As expected, we could not detect CD4[+]Foxp3[+] cells in the DT-treated FTR mouse BM (Supplementary Fig. 1B). Moreover, we observed elevated expression of known T-cell activation markers in DT-treated animals such as CD69 in both CD4[+] and CD8[+] cells and Lag3 in CD4[+] T cells (Supplementary Fig. 1B,C), suggesting that BM T cells acquired an activated phenotype after Treg depletion. Indeed, the serum levels of inflammatory cytokines such as IL-1β, TNFα and IFN-γ were elevated in these conditions (Supplementary Fig. 1D–G). Furthermore, we confirmed that CD4[+] T cells and B220[+] B cells were diffusely located in the BM of untreated mice, while B220[+] B cells could not be found after Treg depletion by immunocytochemistry analysis (Supplementary Fig. 1H). BM B lineage derived (B220[+]) apoptotic cell (annexin-V[+]PI[−]) frequencies of B220[+] cells from DT-treated FTR mice were comparable to those from untreated FTR mice ($P > 0.05$, Student's *t*-test, Supplementary Fig. 1I), indicating that the B cell defect in Treg-depleted animals is not due to an increase in B cell apoptosis.

To evaluate the differentiation capacity of the CD34[−]LSK HSC population after Treg depletion, single-cell colony assays using FACS-sorted CD34[−]LSK HSC derived from WT or FTR mice were analysed with or without DT treatment. The frequencies of colony formation from single-phenotypic HSCs derived from Treg-depleted mice were significantly decreased compared with those from untreated FTR mice or from WT mice (86.8 versus 64.5%, $P < 0.01$, Student's *t*-test, Supplementary Fig. 1J). We also performed competitive repopulation experiments to evaluate if the absence of Treg cells modifies the reconstitution capacity of phenotypic HSC derived from DT-treated or untreated FTR mice *in vivo*. On day 28 after transplantation, the phenotypic HSC derived from Treg-depleted mice showed significantly lower reconstitution in all the lineages analysed (B cells, CD4[+] T cells, CD8[+] T cells and Gr1[+]Mac1[+] myeloid cells, Supplementary Fig. 1K–O). These data suggest that the expanded phenotypic HSCs after Treg depletion had reduced reconstitution capacity on a per-cell basis.

**A normal BM environment rescues the B lymphopoiesis defect.** To estimate the number of functional HSC in total BM cells derived from Treg-depleted mice, we next evaluated the differentiation capacity of total BM cells derived from FTR mice with or without Treg depletion after *in vitro* culture and we found that the frequencies of multi-lineage mixed colony (neutrophil, macrophages, erythrocytes and megakaryocytes) from total BM cells were comparable (Fig. 2a).

We thus performed competitive repopulation experiments to evaluate if a WT BM environment can rescue the lymphoid

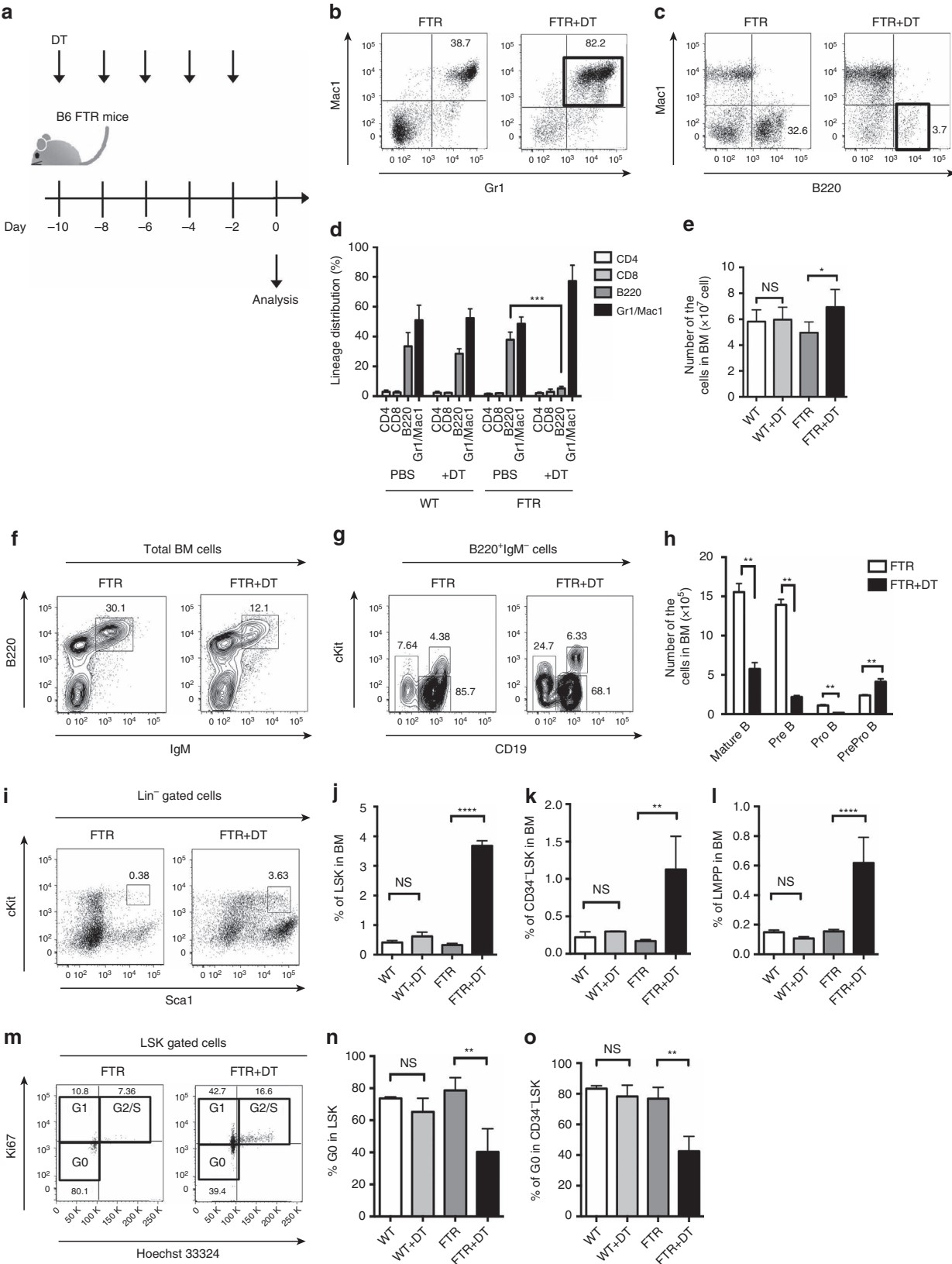

differentiation defect that follows Treg depletion. When comparing the reconstitution capacity of total BM cells from FTR mice with or without DT treatment with WT mice, we found that total BM cells derived from DT-treated FTR mice was still capable of engrafting (Fig. 2b–g). These results demonstrate that the number of functional HSC in total BM cells from Treg-depleted mice was not significantly changed, although the CD34$^-$ LSK population was expanded. Further, the components of the normal BM environment can rescue the defective lymphoid differentiation induced by Treg depletion.

**Host-Treg depletion delays donor B-cell reconstitution.** To investigate the impact of the BM environment after Treg depletion, we next performed transplant experiments using Treg-depleted mice as recipients. Treg depletion was performed by DT treatment on day $-2$ and $-1$ before transplantation followed by lethal irradiation (9.8 Gy) and injection of donor CD45.1 congenic B6 T-cell depleted BM (TCD BM) cells ($1 \times 10^6$ per mouse) on day 0 when Treg cells were not detected in the periphery of transplanted mice (Fig. 3a and Supplementary Fig. 4A). While we could not detect any difference in total peripheral blood (PB) donor chimerism (Fig. 3b,h), Treg-depleted transplanted mice had significantly lower frequencies and absolute numbers of donor B cells compared to untreated animals at various time points following transplantation (Fig. 3c,d,i). Moreover, host-type CD4$^+$ T cells persisted longer in Treg-depleted mice leading to prolonged CD4$^+$ T-cell mixed chimerism (Fig. 3e,j). No clear differences could be seen analysing CD8$^+$ T cell and myeloid cell populations (Fig. 3f,g,k,l).

To investigate B-cell differentiation in these conditions, we further analysed B-cell progenitor cells in the BM at day 28 and 42 after transplantation. The frequencies of donor Pro-B cells and mature B cells were significantly reduced in Treg-depleted mice, while donor-derived LSK and LMPP frequencies were instead increased (Supplementary Fig. 2A–D). Therefore donor B-cell differentiation is impaired in the absence of Treg cells and blocked at early stages of maturation. Moreover, we could not observe differences in production of serum inflammatory cytokines suggesting that the B cell defect is not due to modification of inflammatory cytokine milieu in these conditions (Supplementary Fig. 2E–H).

Next, to further clarify whether delayed immune reconstitution after transplant is directly correlated with Treg depletion and to exclude any possible DT-mediated effects on the BM environment, we transplanted Treg-depleted mice (CD45.2$^+$, FTR mice) with syngeneic TCD BM (CD45.1$^+$ WT-B6 mice) together with $1 \times 10^6$ in vitro activated CD4$^+$CD25$^+$Foxp3$^+$ Treg cells derived from CD45.2$^+$ WT-B6 mice (Fig. 3m,n). Treg cells completely rescued donor B-cell reconstitution in Treg-depleted animals where donor B-cell frequencies after Treg cells transfer were comparable to non-Treg-depleted mice (Fig. 3n). These

results confirm that Treg cells are required for effective donor B-cell reconstitution.

**Treg cells are required for allogeneic donor engraftment.** In a model of allogeneic transplantation where irradiated recipient mice received allogeneic TCD BM, residual host-Treg cells could be detected for at least 4 weeks in spleen, lymph nodes and BM even after lethal irradiation (Supplementary Fig. 3A–E). We also found that host-Treg cells re-isolated 2 weeks after transplantation were functional and could suppress T-cell proliferation in response to an allogeneic stimulus (Supplementary Fig. 3F).

To investigate the role of residual host-Treg cells in vivo after allogeneic transplantation, Treg-depleted FTR mice (H-2$^b$) were lethally (TBI 10 Gy) irradiated and transplanted with TCD BM from allogeneic MHC-major mismatched donor BALB/c (H-2$^d$) mice (Fig. 4a). Treg cells were depleted with DT (Supplementary Fig. 4A). As expected, the transplanted mice did not show any signs of acute GVHD because the donor-derived T cells were removed from the graft[3]. However, around 50% of the Treg-depleted FTR host mice died due to BM aplasia as demonstrated by haematoxylin and eosin staining of the BM collected at day 28 after transplantation (Fig. 4b,c). Moreover, surviving Treg-depleted animals had a significantly lower donor chimerism in comparison to non-Treg-depleted mice (Fig. 4d). The frequencies of donor-derived B cells were significantly lower in Treg-depleted mice, even if the donor graft was not B-cell-depleted and contained ample B cells (Fig. 4e). We also observed reduced donor CD4$^+$, and CD8$^+$ T cell, and myeloid chimerism following Treg depletion (Fig. 4f–h) suggesting that host Treg cells are protective against host-versus-graft BM alloreactivity. To confirm these results, we performed rescue experiments by injecting in vitro activated host-Treg cells in transplanted Treg-depleted animals. Treg-depleted mice that received a Treg cells inoculum ($1 \times 10^6$ per mouse) engrafted and survived similar to non-Treg-depleted control animals (Fig. 4i–k). Strikingly, donor chimerism and donor-derived B-cell frequencies were fully rescued by adoptive transfer of Treg cells and were comparable to untreated control mice (Fig. 4j,k, Supplementary Fig. 4B,C), confirming that host-Treg cells facilitate donor engraftment and immune reconstitution after allogeneic transplantation.

We also evaluated the impact of Treg depletion using purified allogeneic HSC/HPC (haematopoietic progenitor cells) as the stem cell source[10]. A total of $1 \times 10^4$ allogeneic purified LSK from WT-FVB mice (CD45.1, H-2$^q$) were injected into lethally irradiated FTR mice (H-2$^b$) with or without Treg depletion (Supplementary Fig. 4D–J). As expected, donor lymphoid reconstitution was significantly delayed after transplantation (Supplementary Fig. 4E–G). Foxp3$^+$ T-cell frequencies in DT-treated FTR mice were significantly lower than those in non-Treg-depleted mice, but they were eventually recovered by the appearance of donor-derived

---

**Figure 1 | Treg depletion impairs B-cell differentiation in BM.** (**a**) Experimental scheme. (**b,c**) The gating strategy and representative FACS data of total BM cells from FTR mice with or without DT treatment. Note that the frequencies of Gr1$^+$Mac1$^+$ cells were significantly increased in DT-treated FTR mice (**b**) and B220$^+$ cells were decreased (**c**). (**d**) Frequencies of CD4$^+$ T cells, CD8$^+$ T cells, B220$^+$ B cells and Gr1$^+$Mac1$^+$ cells in BM from WT/FTR mice with or without DT treatment ($n = 4$). Data are shown as mean ± s.d. (**e**) Absolute number of total BM cells in WT/FTR mice with or without DT treatment. ***$P < 0.001$, Student's $t$-test. Data are shown as means ± s.d. (**f,g**) Representative results of B-cell progenitor analysis in BM from FTR mice with or without DT treatment. Typical plots of IgM$^+$B220$^+$ mature B cells (**f**), IgM$^-$B220$^+$CD19$^+$cKit$^-$ Pre-B cells, and IgM$^-$B220$^+$CD19$^+$cKit$^+$ Pro-B cells are shown (**g**). (**h**) The graph shows the total number of B-cell progenitors in BM derived from the FTR mice with or without DT treatment ($n = 5$, **$P < 0.01$, Student's $t$-test). Data are shown as means ± s.d. (**i**) Representative results of HSC analysis in BM from FTR mice with or without DT treatment. Note the significant increase of LSK population and CD34$^-$ LSK population in FTR mice with DT treatment. (**j–l**), Frequencies of LSK (**j**), HSC (CD34$^-$LSK; (**k**)), and LMPP (Flt3$^+$LSK; (**l**)) in BM ($n = 4$ in each group, **$P < 0.01$, ****$P < 0.0001$, Student's $t$-test) are also shown. Data are shown as means ± s.d. (**m–o**) The cell cycle status in HSCs derived from FTR mice with or without DT treatment is reported. Representative dot plot data (**m**), frequencies of G0 state in LSK (**n**) and in CD34$^-$LSK (**o**) are shown ($n = 4$ in total, **$P < 0.01$, Student's $t$-test). Data are shown as means ± s.d.

Treg cells 12 weeks after transplantation and simultaneously donor-derived B cells gradually recovered (Supplementary Fig. 4G,I,J). Because mature donor cells including mature B cells and T cells were not included in the donor graft, these data suggested that Treg depletion can modify the BM environment for B-cell differentiation from HSC/HPC.

**Host-Treg cells adoptive transfer promotes B-cell reconstitution.** We next examined whether Treg cells adoptive transfer could promote B-cell reconstitution in a mouse transplant model using immune-deficient mice where host-versus-graft immune reactions are absent or minimal (Fig. 5). A total of $1 \times 10^6$ allogeneic TCD BM cells from WT-B6 mice were injected into

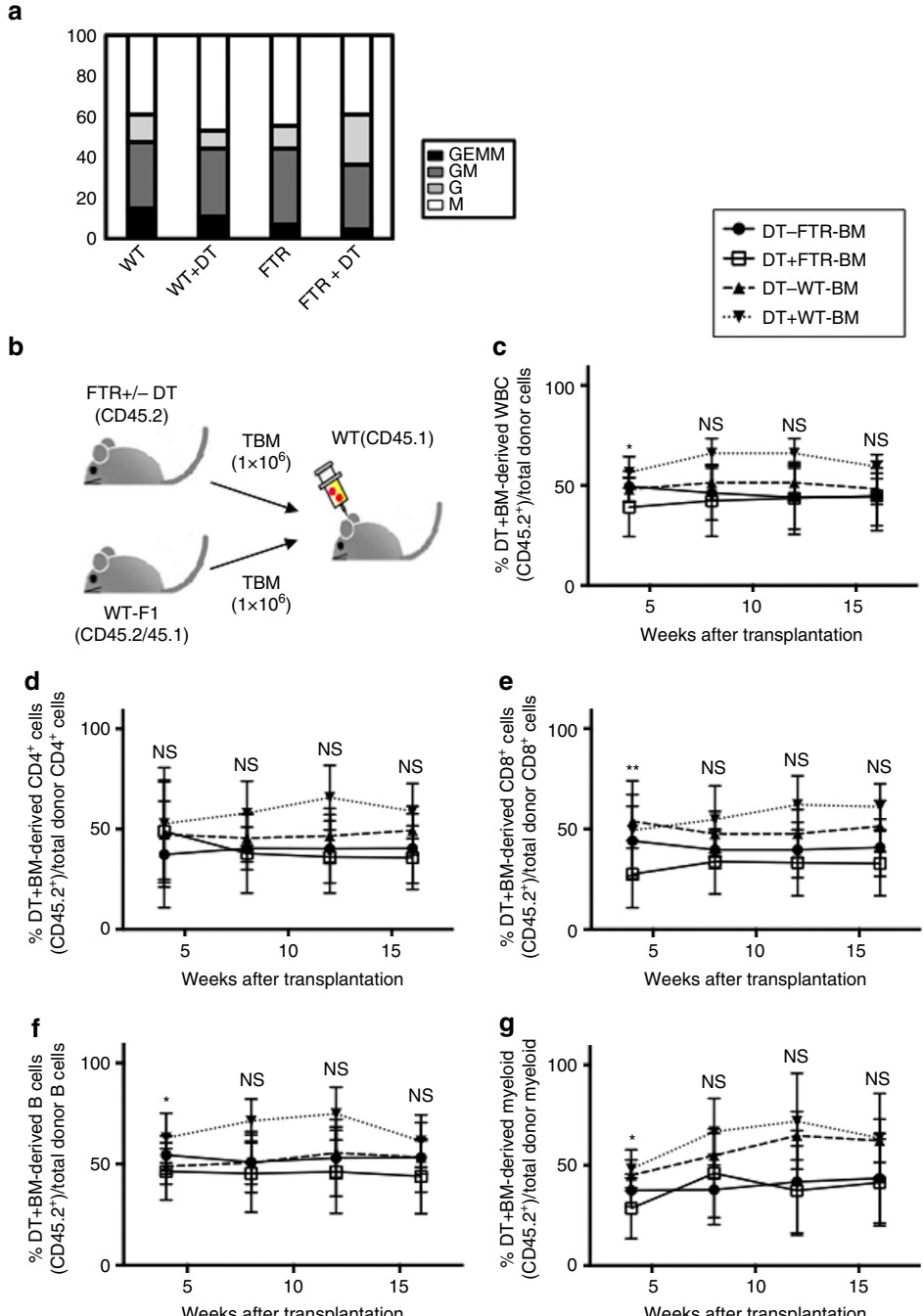

**Figure 2 | Normal BM environment rescues B cell defect.** (**a**) Results of methylcellulose colony assay using total BM cells from Treg-depleted or not FTR mice are shown. Colony distribution is reported. GEMM; granulocyte, erythrocyte, macrophage, GM; granulocyte, monocytes, G; granulocyte, M macrophage. Pooled data from three consecutive experiments are shown. (**b**) Experimental scheme of competitive repopulation assay using $1 \times 10^6$ total BM cells from FTR mice (CD45.2) with or without DT treatment. As competitor cells, the same number of total BM cells from WT-F1 B6 mice (CD45.1/CD45.2) was co-injected into lethally irradiated CD45.1 B6 mice. (**c–g**) The frequencies of CD45$^+$ (**c**), CD4$^+$ T cells (**d**), CD8$^+$ T cells (**e**), B220$^+$ B cells (**f**), Gr1$^+$Mac1$^+$ granulocytes (**g**) derived from FTR mice at 4, 8, 12, 16 weeks after transplant are reported. ns = not significant, DT-FTR-BM versus DT + FTR-BM on 4 weeks after transplant in (**c**), (**f**), (**g**); *$P < 0.05$, Student's $t$-test, DT-FTR-BM versus DT + FTR-BM on 4 weeks after transplant in (**e**). **$P < 0.01$, Student's $t$-test. Data are shown as means ± s.d.

non-irradiated BALB/c-rag2$^{-/-}$γc$^{-/-}$ mice with or without *in vitro* activated BALB/c-derived Treg cells. We observed donor engraftment including CD4$^+$, CD8$^+$ T cells, even when we used TCD-BM. The adoptive transfer of host-type Treg cells enhanced donor chimerism over time and boosted donor B cell reconstitution as detected in the PB, BM and the spleen, while CD4$^+$

and CD8$^+$ T cell frequencies did not show significant difference between the groups (Fig. 5a–e).

Moreover, donor-derived CD8$^+$ T cells in the BM of Treg-treated animals had reduced expression of activation markers such as CD69, CD44 and PD1 (Fig. 5f) and we detected reduced amounts of inflammatory cytokines such as IL-1β, IFNγ

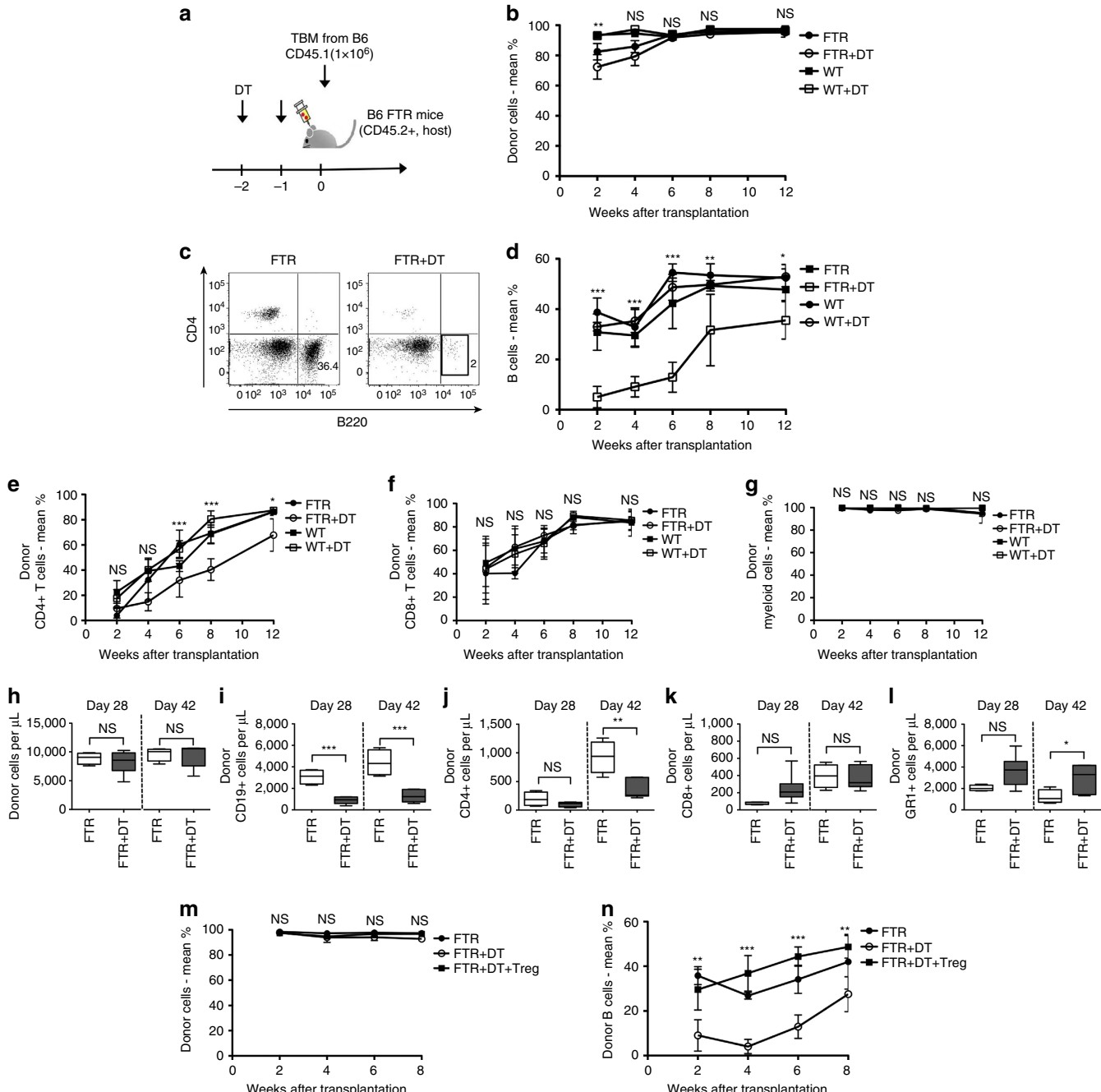

**Figure 3 | Host-Treg depletion delays donor B cell reconstitution. (a)** Experimental scheme: FTR mice (CD45.2) with or without DT treatment on Day -2 and -1 were lethally irradiated and transplanted with $1 \times 10^6$ total BM cells from congenic CD45.1$^+$ B6 mice. **(b)** Graph of PB donor chimerism of total CD45$^+$ cells over time after transplantation of DT-treated and untreated FTR mice. **(c)** Representative data of CD4$^+$ T cells and B220$^+$ B cells in PB at day 28 after transplantation from Treg-depleted (right) or not (left) mice. **(d)** PB donor B cell percentage over time after transplantation of DT-treated and untreated FTR mice. **(e–g)**. PB donor chimerism of **(e)** CD4$^+$ T cells, **(f)** CD8$^+$ T cells, and **(g)** myeloid cells over time after transplantation of DT-treated and untreated FTR mice. **(h–l)**. Absolute number at day 28 and day 42 after transplantation of **(h)** total CD45$^+$ cells, **(i)** B cells, **(j)** CD4$^+$ T cells, **(k)** CD8$^+$ T cells and **(l)** myeloid cells. **(m,n)** Total CD45$^+$ cell donor chimerism **(m)** and donor B cell percentage **(n)** in PB of untreated FTR mice, DT-treated FTR mice and DT-treated FTR mice that were rescue with Treg cells infusion at day 0 of transplantation are shown. ns = not significant, *$P < 0.05$, **$P < 0.01$, ***$P < 0.001$, Student's *t*-test. Data are shown as means ± s.d.

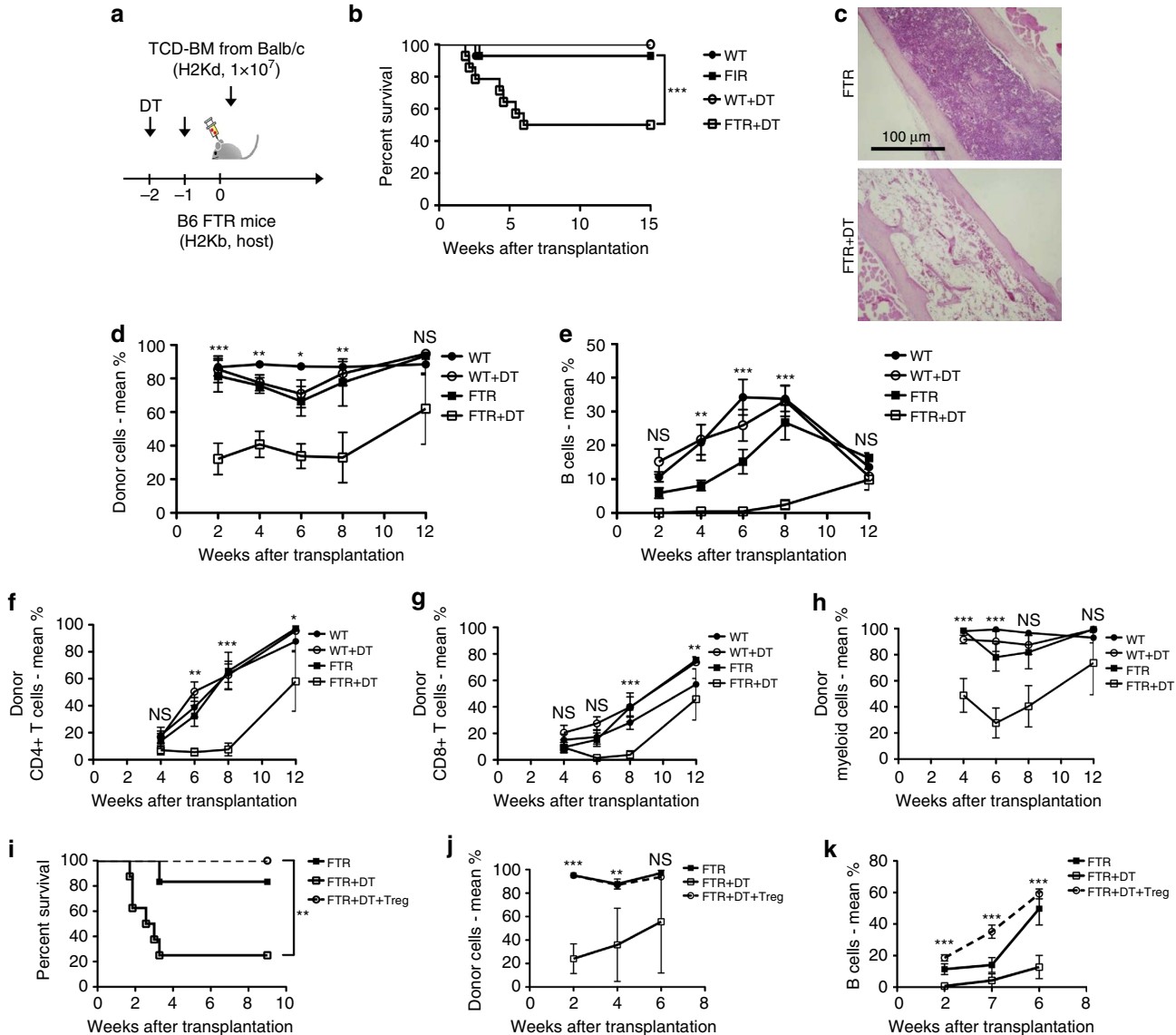

**Figure 4 | Allogeneic donor engraftment is impaired by host-Treg-depletion.** (**a**) Experimental scheme: B6 WT or B6 FTR (both H-2[b]) mice with or without DT treatment on day − 2 and − 1 were lethally irradiated and transplanted with $1 \times 10^7$ TCD BM from allogeneic BALB/c (H-2[d]) mice. (**b**) Survival of transplanted mice. Comparisons have been made between DT-untreated and DT-treated B6 WT and B6 FTR mice. The graph shows the results of three-pooled consecutive experiments (***$P < 0.001$, log-rank test). (**c**) Representative histological haematoxylin and eosin stained section of BM derived from allogeneic transplanted untreated or DT-treated FTR mice 4 weeks after transplantation. (**d**) PB donor chimerism of total CD45[+] cells over time after transplantation of DT-treated and untreated B6 WT or B6 FTR mice. Statistical analysis of the differences between FTR and DT-treated FTR at different time points are shown. (**e**) PB donor B-cell percentage over time after transplantation of DT treated and untreated B6 WT or B6 FTR mice. Statistical analysis of the differences between FTR and DT-treated FTR at different time points are shown. (**f**–**h**). PB donor chimerism of CD4[+] T cells (**f**), CD8[+] T cells (**g**), and Myeloid cells (**h**) over time after transplantation of DT-treated and untreated B6 WT or B6 FTR mice. Statistical analysis of the differences between FTR and DT-treated FTR at different time points are shown. Data are shown as means ± s.d. ns = not significant, *$P < 0.05$, **$P < 0.01$, ***$P < 0.001$, Student's $t$-test. (**i**) Survival of transplanted mice when host-Treg cells have been adoptively transferred to rescue donor BM engraftment. Comparisons have been made between DT-untreated FTR mice, DT-treated FTR mice and DT-treated FTR mice that received host-Treg cells infusion at day 0 of transplantation. Data reported is the results of two-pooled consecutive experiments. **$P < 0.01$, log-rank test. (**j**,**k**) Total CD45[+] cell donor chimerism (**j**) and donor B-cell percentage (**k**) in PB of untreated FTR mice, DT-treated FTR mice and DT-treated FTR mice that were rescue with host-Treg cells infusion at day 0 of transplantation are shown. Statistical analysis of the differences between DT-treated FTR and DT-treated FTR rescued with host-Treg cells at different time points are shown. Data are shown as means ± s.d. ns = not significant, *$P < 0.05$, **$P < 0.01$, ***$P < 0.001$, Student's $t$-test.

and TNFα in their sera (Fig. 5g,h), demonstrating that host-type Treg cells control donor T-cell activation and production of inflammatory cytokines favouring donor engraftment and B-cell immune reconstitution.

To confirm these data and to provide a tool for further clinical translation in immune-competent hosts, we transplanted

$luc^+$ B6 TCD BM into sublethally irradiated (TBI 5.5 Gy) BALB/c mice and observed the impact of host-Treg cells adoptive transfer ($5 \times 10^5$ per mouse) on donor engraftment and immune reconstitution (Supplementary Fig. 5). Mice received intraperitoneal injections of low-dose IL-2 for 7 days to allow for Treg cells *in vivo* activation. PB chimerism analysis

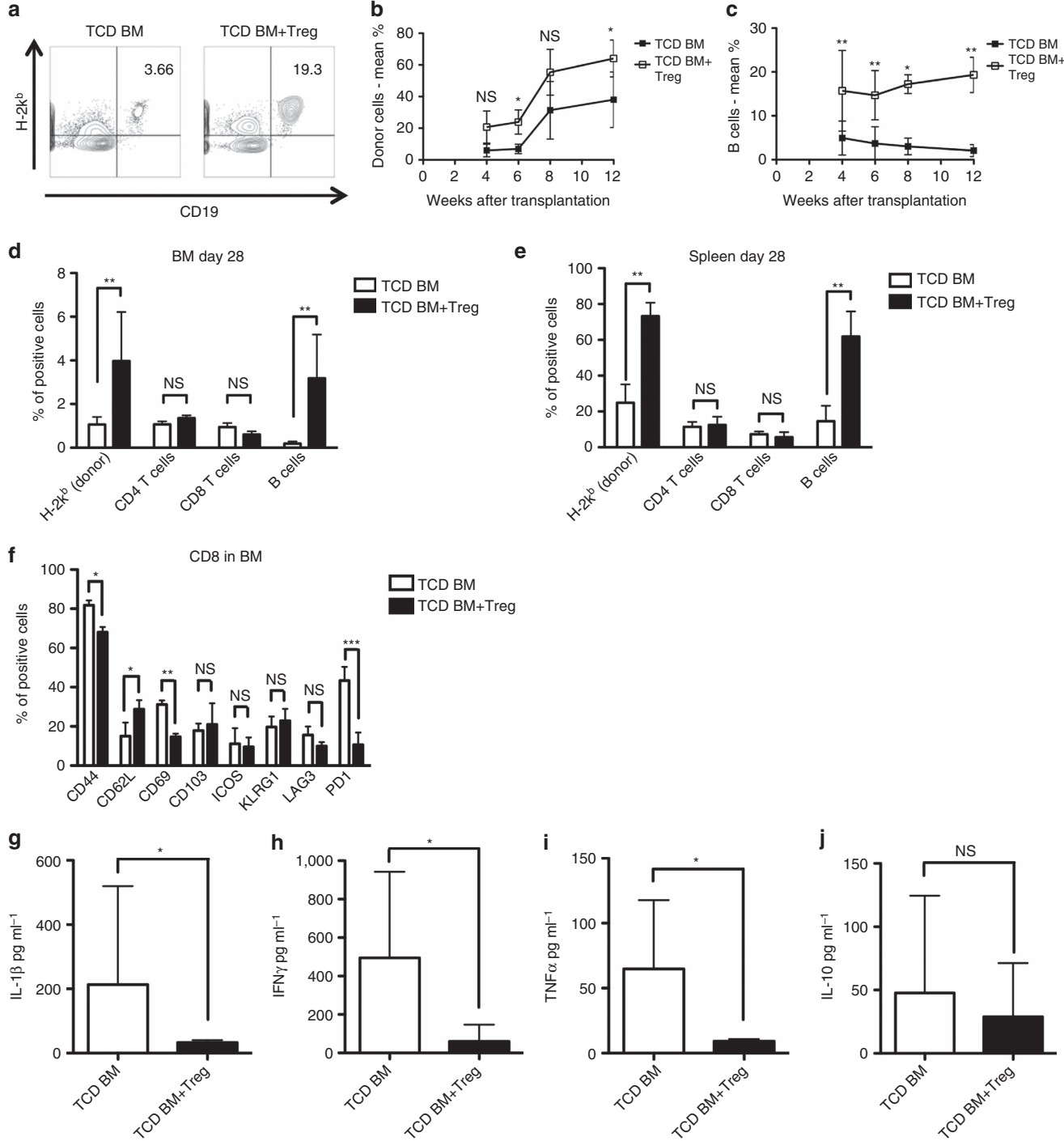

**Figure 5 | Host-Treg cells adoptive transfer promotes B-cell reconstitution.** Immune-deficient BALB/c-rag2$^{-/-}$γc$^{-/-}$ (H-2$^d$) mice were transplanted with $1 \times 10^6$ TCD BM from allogeneic B6 (H-2$^b$) mice and were treated with host-Treg cells at day 0 of transplantation. (**a**) Donor H-2k$^b$+CD19$^+$ B cells in PB of transplanted BALB/c-rag2$^{-/-}$γc$^{-/-}$ mice in the absence (left) or presence (right) of host-Treg cells treatment. (**b,c**) PB donor chimerism of total CD45$^+$ cells (**b**) and donor B-cell percentage (**c**) over time after transplantation of Treg cells-treated and untreated BALB/c-rag2$^{-/-}$γc$^{-/-}$ mice. Data are shown as means ± s.d. ns = not significant, *$P < 0.05$, **$P < 0.01$, Student's $t$-test. (**d,e**) Frequencies of total donor cells, donor CD4$^+$ T cells, donor CD8$^+$ T cells and donor B cells in the BM (**d**) and in the spleen (**e**) of transplanted mice at day 28 after transplantation ($n = 5$ in each group). Data are shown as means ± s.d. ns = not significant, **$P < 0.01$, Student's $t$-test. (**f**) Frequencies of CD44, CD62L, CD69, CD103, ICOS, KLRG1, LAG3 and PD1 expressions on donor CD8$^+$ T cells in the BM of transplanted mice at day 8 after transplantation ($n = 5$ in each group). Data are shown as means ± s.d. ns = not significant, *$P < 0.05$, **$P < 0.01$, ***$P < 0.001$, Student's $t$-test. (**g-j**). Serum concentrations of different cytokines in transplanted BALB/c-rag2$^{-/-}$γc$^{-/-}$ mice that received or not host-Treg cells adoptive transfer. Data are shown as means ± s.d. ns = not significant, *$P < 0.05$, Student's $t$-test.

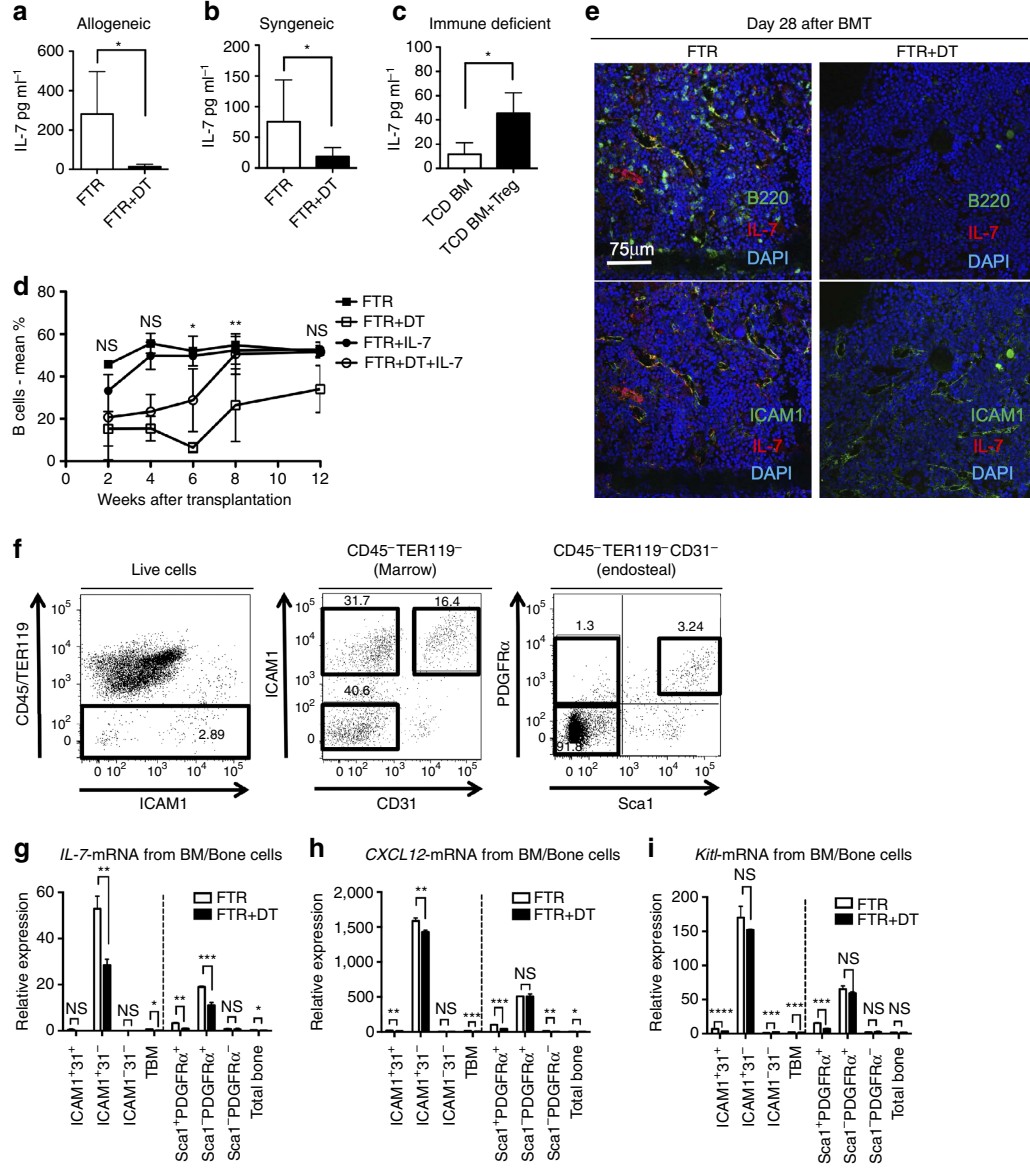

**Figure 6 | Treg-depletion reduces IL-7 production from ICAM1$^+$ stroma. (a–c)** Graphs show IL-7 concentrations in the serum of Treg-depleted mice after syngeneic transplant (**a**), after allogeneic transplant (**b**), and in allogeneic transplanted BALB/c-rag2$^{-/-}$γc$^{-/-}$ mice (**c**) that received or not host-Treg cells adoptive transfer ($n=5$ per group). Data are shown as means ± s.d. *$P<0.05$, Student's $t$-test. (**d**) Donor B-cell percentage in PB of untreated or DT-treated FTR mice that received intraperitoneal administration of low-doses of IL-7. Statistical analysis of the differences between DT-treated FTR that received IL-7 and DT-treated FTR alone at different time points are shown. Representative data from one of three experiments is shown. Data are shown as means ± s.d. ns = not significant, *$P<0.05$, **$P<0.01$, Student's $t$-test. (**e**). Representative results of immunostaining using Treg-depleted FTR mice 28 days after syngeneic transplantation (right panels). The images of transplanted untreated FTR mice are also shown as control (left panels). Note that the frequencies of B220$^+$ cells (upper panels) and IL-7$^+$ cells (upper and lower panels) are severely decreased in Treg-depleted FTR mice. (**f**). Representative plot data of non-haematopoietic cells in BM from WT mice. Total BM cells were flushed out and digested with collagenase. Bone related cells were also isolated from bone-derived digested cells. After gating for CD45$^-$TER119$^-$ cells (left panel), digested marrow cells were gated for ICAM1 and CD31 expression (middle) and bone related cells were gated for Sca1 and PDGFRα expression (right). (**g–i**). Quantitative RT-PCR data using non-haematopoietic cells from untreated or DT-treated FTR transplanted mice. il-7 mRNA (**g**), cxcl12 mRNA (**h**), and kitl mRNA (**i**) are shown ($n=3$ in each group). Data are shown as means ± s.d. ns = not significant, *$P<0.05$, **$P<0.01$, ***$P<0.001$, Student's $t$-test.

and *in vivo* imaging showed transient donor engraftment followed by rapid rejection within 2 weeks after transplantation in mice that received only donor TCD BM. Mice that received TCD BM + host-Treg cells alone or TCD BM + IL-2 alone did not have a significant improvement in donor engraftment, while mice that received TCD BM + both host-Treg cells and IL-2 treatment had an increased and persistent donor engraftment (Supplementary Fig. 5A,F,G).

All the donor-derived white cell lineage frequencies (CD4$^+$ T cells, CD8$^+$ T cells, B cells and myeloid cells) were strikingly increased by host-Treg cells + IL-2 treatment, but while donor myeloid cells were no more detectable after 4 weeks from transplantation, host-Treg cells + IL-2 treatment ensured a durable and persistent (analysis up to 120 days after transplantation) donor lymphoid chimerism (Supplementary Fig. 5B–E).

**Treg depletion reduces IL-7 production by ICAM1$^+$ stroma.** For B-cell differentiation from HSC, extrinsic humoural factors such as IL-7 or C-X-C motif chemokine 12 (CXCL12, also known as stromal cell derived factor-1; SDF-1) are required[11]. These cytokines are thought to be secreted by non-haematopoietic stromal/niche cells in the BM. According to these reports, we hypothesized that the B-cell differentiation defect after Treg depletion could be due to a dysfunction of the cytokine-secreting BM niche cells.

Indeed, in the sera obtained from Treg-depleted transplanted mice IL-7 levels were lower compared with those without Treg depletion (Fig. 6a–c). To further prove a role for Treg cells in modulating IL-7 production *in vivo*, we transplanted lethally irradiated Treg-depleted FTR mice with TCD BM from congenic CD45.1$^+$ B6 mice and low-dose IL-7 (1 µg per mouse per day) for 7 consecutive days. As expected, all mice engrafted and no differences were found in total donor cells, CD4$^+$ or CD8$^+$ T cell chimerism (Supplementary Fig. 6A–D). IL-7 treatment almost completely rescued donor B cell reconstitution in Treg-depleted mice by 8 weeks after transplant (Fig. 6d, Supplementary Fig. 6E). IL-7 treatment without Treg depletion did not induce modifications in B cell numbers and reconstitution suggesting that physiological levels of IL-7 are sufficient for donor B cell differentiation. These data demonstrate that Treg cells promote B-cell differentiation and reconstitution through an IL-7 dependent mechanism. IL-7 expressing cells in BM have been examined by immunohistochemistry or il-7-reporter mice and IL-7 is produced by BM stromal cells[8], suggesting that IL-7 is not directly secreted by Treg cells.

Intercellular Adhesion Molecule 1 (ICAM1, also known as CD54) is a surface protein, which is mainly expressed on activated T cells, endothelial cells and stromal cells in the BM. ICAM1$^+$ non-haematopoietic/endothelial cells were reported to be IL-7 and CXCL12 secreting cells and are thought to play an important role in B-cell differentiation[12]. Indeed, immunofluorescence analysis using fresh BM showed that ICAM1$^+$ VE-Cadherin$^-$ cells were located in the perivascular areas and were positive for IL-7, although the majority of ICAM1$^+$ VE-Cadherin$^+$ endothelial cells were IL-7 negative (Supplementary Fig. 7A,B). Strikingly, IL-7$^+$ICAM1$^+$ cells were hardly detected in the Treg-depleted host mice (Day 28, Fig. 6e). Quantitative RT-PCR analysis showed that FACS-sorted BM ICAM1$^+$CD31$^-$CD45$^-$TER119$^-$ stromal cells showed higher levels of il-7 and cxcl12 mRNA compared with ICAM1$^+$CD31$^+$CD45$^-$TER119$^-$ endothelial cells, or total BM samples (Fig. 6f–i). Importantly, il-7 expression was hardly detected in all the other analysed populations (ICAM1$^+$CD31$^+$ CD45$^-$TER119$^-$ endothelial cells, ICAM1$^-$CD31$^-$CD45$^-$ TER119$^-$ non-haematopoietic cells, total BM samples, Sca1$^+$ PDGFRα$^+$ MSC, or Sca1$^-$PDGFRα$^-$ endosteal stromal cells from digested bone cells). Therefore, the major source of IL-7 in the BM is represented by ICAM1$^+$CD31$^-$CD45$^-$TER119$^-$ stromal cells. This population is also positive in stem cell factor (SCF, the ligand for cKit), which plays a crucial role for HSC maintenance and B cell differentiation[11], when analysed in SCF-GFP mice (Supplementary Fig. 7C–F).

We next analysed *il-7, cxcl12* and *kitl (ckit ligand)* expression levels of ICAM1$^+$CD31$^-$CD45$^-$TER119$^-$ stromal cells in BM derived from transplanted mice with or without Treg depletion (Fig. 6g–i). However, the expression level of *il-7* in ICAM1$^+$ CD31$^-$CD45$^-$TER119$^-$ stromal cells from Treg-depleted transplanted mice was significantly lower in comparison to non-Treg-depleted mice (Fig. 6g). *cxcl12* and *kitl* were also abundantly expressed in ICAM1$^+$CD31$^-$CD45$^-$TER119$^-$ stromal cells but were less impacted by Treg depletion (Fig. 6h,i).

These data provide evidence that the delayed immune reconstitution including B cell differentiation from HSC after

Treg depletion was induced by the dysfunction of IL-7 secreting ICAM1$^+$CD31$^-$CD45$^-$TER119$^-$ perivascular cells in the BM. Treg cells not only provide an immunological barrier to the HSC-niche[6], but also promote their further lymphoid differentiation through control of IL-7 production via ICAM1$^+$ CD31$^-$CD45$^-$TER119$^-$ perivascular cells. The presence of activated T cells and increased levels of inflammatory cytokines in the BM in the absence of Treg cells suggest that Treg cells provide immune protection to HSC and IL-7-secreting ICAM1$^+$CD31$^-$ CD45$^-$TER119$^-$ perivascular cells. Treg cells are also required for controlling BM immune homeostasis and development/ differentiation in physiologic conditions and after transplantation (proposed model, Fig. 7).

## Discussion

Foxp3$^+$ Treg cells are known to promote immunological tolerance and suppress autoimmunity. Here, we show that Treg cells are also required to maintain BM stromal cell function and that these cells are necessary for immune reconstitution after transplantation. We demonstrate that Treg cells are required for B-cell differentiation under physiologic conditions and after syngeneic and allogeneic transplantation. ICAM1$^+$CD31$^-$ CD45$^-$TER119$^-$ perivascular cells were shown to be the main producers of IL-7 and are a possible target of self-activating T cells after Treg depletion.

Treg cells have been shown to possess tolerogenic properties in several transplantation models[2]. One of the main mechanisms of Treg cells function is thought to be their ability to suppress the proliferation and activation of T cells that mediate donor HSC rejection. Indeed, the adoptive transfer of Treg cells rescued the engraftment failure that was induced in allogeneic transplant using Treg-depleted recipient mice (Fig. 4). Inappropriate inflammatory cytokine signalling, such as IFN or TNF can induce HSC dysfunction[13,14]. These cytokine signals might be key regulators in the Treg-depleted transplant model (Supplementary Fig. 1). However, the number of functional HSC was not significantly changed in transient Treg depletion model (Figs 1 and 2, Supplementary Fig. 1), although the phenotypic HSC population entered the cell cycle and expanded (Fig. 1). Instead, a defect of B-cell differentiation was observed (Fig. 1) upon host-Treg depletion suggesting that the BM environment for B-cell differentiation was more susceptible to immune cell activation after Treg depletion (Figs 3 and 4). These findings imply that Treg cells might regulate not only inflammatory cytokine secretion which controls HSC quiescence but also the function of the BM environment. Moreover, Treg depletion results in engraftment failure and impacts on all the lineages when mice are transplanted in allogeneic conditions suggesting that HSC and BM environment have a different degree of susceptibility to the reactive host immune cells.

The presence of a cytokine pool is critical for lymphoid differentiation from HSC in each differentiation stage[11]. IL-7 and CXCL12 are critical for differentiation into B-cell lineages. Exogenous IL-7 treatment of transplanted Treg-depleted mice could rescue donor B-cell reconstitution, suggesting that Treg cells might control IL-7 production. IL-7 has been used to boost immune reconstitution after transplantation in preclinical studies and in clinical trials[15,16]. Exogenous IL-7 injected into syngeneic transplanted mice that were not Treg-depleted did not result in enhanced immune reconstitution in comparison to untreated animals confirming that in our model there was sufficient physiological IL-7 production and that only Treg-depleted mice lacked IL-7.

Osteoblasts were reported as a possible source of IL-7 (ref. 17). Others, through the use of IL-7 promoter GFP mice, suggested

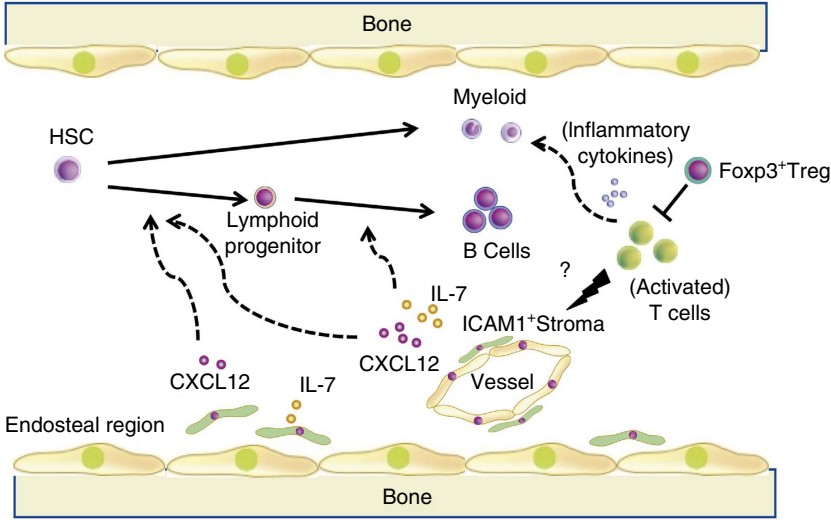

**Figure 7 | Schematic model of the role of Treg cells in BM microenvironment.** Foxp3[+] Treg cells in BM regulate the activation status of cytotoxic T cells and maintain the function of BM environment including ICAM1[+] perivascular cells. Activated T cells after Treg cells depletion could abrogate IL-7 secretion from ICAM1[+] perivascular cells resulting in a defective B-cell differentiation from HSC.

that BM reticular stromal cells also secrete IL-7 (ref. 11). In our qRT-PCR studies, *il-7* mRNA was much higher in ICAM1[+] CD31[−] perivascular cells than those in Sca1[−]PDGFRα[+] endosteal cells, which are derived from bone digested samples. These data indicate that the major source of IL-7 in BM is represented by ICAM1[+] perivascular stromal cells, which retain the characteristics of osteoblast progenitors, a progeny of MSC[18]. In a recent study, infection-related cytotoxic T cells stimulate interleukin-6 (IL-6) secretion from MSC and promote myelopoiesis[19]. Our data show that IL-7 secreting ICAM1[+] perivascular stromal cells represent another target of self-activated T cells after Treg depletion.

Recently, extensive chronic GVHD after human transplantation was associated with osteoblast and B cell loss[20]. In concordance with such a hypothesis, we also found decreased levels of *il-7* mRNA in Sca1[−]PDGFRα[+] bone associated cells after Treg depletion (Fig. 6e). However, our data imply that activated T cells would not only target osteoblast but also ICAM1[+] perivascular stromal cells. ICAM1 is a ligand for LFA-1, which is expressed on T cells, B cells, macrophages and neutrophils and it is involved in recruitment to the site of immune reactions[21]. This might be the reason why ICAM1[+] perivascular stromal cells would become a preferred target of activating T cells after Treg depletion.

*Foxp3* deficient mice ('scurfy') and patients with defective or absent FOXP3 expression (IPEX, immune dysregulation, polyendocrinopathy, enteropathy, X-linked, syndrome) have high levels of circulating autoantibodies suggesting that the Treg cells defect is also followed by B cell dysfunction[22,23]. It has been also proposed that uncontrolled autoimmune reactions may be responsible for defective B lymphopoiesis in the fetus[24]. However, the role of Treg cells in B cell differentiation from HSC in adult BM and following transplantation could not be fully elucidated since scurfy mice die soon after birth. Our data clearly showed that Treg cells also play a critical role in adult immune reconstitution including B cell differentiation and maintain the function of IL-7 secreting ICAM1[+]CD31[−] stromal cells. Therefore, Treg cells not only regulate the inflammatory cytokine levels in the BM but also the cytokine-secreting stromal cell function. We suggest that Treg cells play a key role in immune-surveillance so that autoimmune like activating

effector T cells cannot abrogate the function of the BM microenvironment.

Treg cells can regulate B cell function through direct B cell killing[25,26] or via CTLA4-mediated interactions[25,26] and by limiting B cell production of autoantibodies[27,28]. We further demonstrate that Treg cells are required for maintaining immune homeostasis at the HSC and B cell precursor niche level, therefore playing a relevant role in B cell differentiation.

Several studies have shown that Treg cells/CD4[+] T cell ratios are higher in the BM than in secondary lymphoid tissues[29,30]. Here we show that BM Treg cells play a key role in normal lymphopoiesis through the protection of BM environment. Our data suggest that the regulation of the activation state of residual BM T cells by Treg cells is a major mechanism for preserving normal haematopoiesis and reconstitution. However, further studies are needed to investigate other possible mechanisms independent on the regulation of effector T cells.

Previous reports demonstrated that T cells are resistant to an otherwise lethal dose of radiation *in vivo*[31–33]. Our results confirm this finding, and further demonstrate that Treg cells have a functional advantage over conventional T cells as they do not require a high proliferation rate to exert their function; therefore the activation of the host T-cell pool that follows irradiation is counterbalanced by Treg cells suppressive activity.

We also showed that Treg cells adoptive transfer ameliorated donor engraftment and boosted immune reconstitution after transplantation in the absence of any treatment in immune-deficient animals and after only sublethal irradiation in immune-competent hosts. These data suggested that activation of donor-derived T cells was regulated by the adoptive Treg cells transfer. Thus, Treg cells-based cellular therapy could represent a safer, less toxic and highly effective approach in transplantation of patients that are unfit for myeloablative conditioning regimens or of patients with non-malignant haematological malignancies such haemoglobinopathies where treatment toxicity is a major concern[34]. Another clinical implication of our data could be transplantation for inherited immune-deficient patients as delayed reconstitution of humoral immunity is one of the major problems in such therapeutic approaches[35,36] and a Treg cells imbalance after transplant could cause delayed B-cell reconstitution.

In summary, we show that Treg cells maintain the function of BM ICAM1$^+$CD31$^-$ stromal cells, and support lymphopoiesis including B-cell differentiation from HSC. These results provide new critical insights into Treg cells biology and function, and suggest further clinical applications for the treatment of delayed immune reconstitution after transplantation.

## Methods

**Mice.** Wild type C57BL/6 (B6, H-2$^b$, CD45.2$^+$), CD45.1$^+$ congenic mice, and BALB/c (H-2$^d$, CD45.2$^+$) mice were purchased from the Jackson Laboratory (Sacramento, CA). B6 FTR (FTR, H-2$^b$, CD45.2$^+$, from Dr Rudensky, New York, USA), BALB/c-rag2$^{-/-}$γc$^{-/-}$ and B6-Foxp3.Luci.DTR-4 (H-2$^b$) mice (from Dr Hammerling, Heidelberg, Germany) were bred in our animal facility at Stanford University. Eight to twelve weeks old gender matched mice were used in the experimental procedures. All animal protocols were approved by the Institutional Animal Care and Use Committee at Stanford University.

**Antibodies and reagents.** Information of all antibodies are shown in Supplementary Table 1. Fixable Viability Dye eFluor 506 (eBioscience) was used to exclude dead cells. FoxP3 Fixation/Permeabilization buffer set was purchased from eBioscience. DT was purchased from Sigma-Aldrich. Treg depletion was obtained as described in (ref. 37). Briefly, in transplantation experiments daily 50 mg kg$^{-1}$ DT was injected intraperitoneally at days $-2$ and $-1$ before transplantation; for Treg depletion in untreated mice 50 mg kg$^{-1}$ DT was injected intraperitoneally every other day for five times. In both the cases efficiency of Treg depletion was checked the day after the last DT injection by PB FACS analysis.

**Purification of HSC, Treg cells and stromal cells.** Mouse CD34 − LSK HSC and CD4$^+$CD25$^+$ Treg cells were isolated as previously described[3,38]. Briefly, for HSC isolation, BM lineage negative cells were enriched by biotinylated anti CD4, CD8, B220, TER119, Gr1, CD127 antibodies, anti biotin beads and LS column. Then, CD34$^-$cKit$^+$Sca1$^+$Lineage$^-$ cells were sorted by FACS. For Treg cells isolation, cells from lymph nodes and spleen were stained with anti-CD25 APC, incubated with anti-APC microbeads and positively selected with a LS column (Miltenyi). CD4$^+$CD25$^+$ cells were sorted and the purity of CD4$^+$CD25$^+$FoxP3$^+$ was >95%.

For the isolation of stromal cells BM cells were digested with 100 U ml$^{-1}$ collagenase IV and 10 U ml$^{-1}$ DNase Invitrogen (I) at 37 °C for 30 min. Residual bone tissues were digested with collagenase to isolate bone related non-haematopoietic cells[39]. Non-haematopoietic cells were enriched by CD45/TER119 negative selection. Cells were sorted on a FACS Aria (BD Bioscience) and all flow cytometry data were analysed with FlowJo software (Tree Star).

**Mixed lymphocyte reactions.** Isolated T cells were incubated with irradiated (32 Gy) allogeneic splenocytes at different ratios as indicated and cultured in the presence of IL-2 (100 IU ml$^{-1}$, Roche). Cell proliferation was assessed through H$^3$-thymidine incorporation[26]. In the experiments with Treg cells co-incubation purified Treg cells were added at an 1:2 Treg cells:T cell ratio.

**Transplantation.** A total of $1 \times 10^6$ BM cells derived from CD45.1$^+$ B6 mice were injected into lethally irradiated FTR mice with or without DT on day $-2$ and $-1$ in syngeneic transplants. For the allogeneic transplant model, $5 \times 10^6$ TCD-BM (the frequencies of residual CD4$+$ and CD8$+$ T cells in the donor graft were below 0.5%) cells from BALB/c (CD45.2, H-2$^d$) mice or $1 \times 10^4$ LSK HSCs derived from WT-FVB mice (CD45.1, H-2$^q$) were injected into lethally irradiated FTR mice with or without DT on day -2, and -1. After transplantation PB-cell chimerism was calculated according to the frequencies of donor type MHC.

In the competitive repopulation assays, chimerism was monitored through the CD45 congenic marker system. A total of $2 \times 10^5$ BM cells or 100 CD34$^-$ LSK cells derived from FTR (CD45.2) with or without Treg depletion were mixed with $2 \times 10^5$ BM competitor cells from B6 F1 mice (CD45.1/CD45.2) and were transplanted into CD45.1$^+$ congenic B6 mice irradiated at a dose of 9.8 Gy. The reconstitution capacity of test cells was calculated as (% of CD45.2$^+$ cells)/(% of CD45.1$^+$/CD45.2$^+$ + CD45.2$^+$ cells) × 100.

In experiments with immune-deficient mice, BALB/c-rag2$^{-/-}$γc$^{-/-}$ mice were transplanted with $1 \times 10^6$ allogeneic B6 TCD BM cells/mouse in the absence of any conditioning and $1 \times 10^6$ in vitro activated host-Treg cells were adoptively transferred on day 0.

In experiments with Treg adoptive transfer in immune-competent hosts, sublethally irradiated (5.5 Gy) BALB/c mice were transplanted with $5 \times 10^6$ allogeneic $luc^+$ B6 TCD BM cells/mouse and treated with or without $5 \times 10^5$ freshly-isolated host-Treg cells with or without intraperitoneal low-dose IL-2 administration (50,000 IU × 2 daily from day 0 to 6). Bioluminescence imaging (BLI) was used to quantify donor engraftment over time as previously reported[40]. Briefly, mice were injected intraperitoneally with luciferin (10 µg g$^{-1}$ of body weight). The mice were anesthetized and imaged using an IVIS Spectrum charge-coupled device imaging system (Caliper-Xenogen) for up to 5 min. Imaging

data were analysed with Living Image Software (Caliper Life Sciences). For IL-7 treatment experiments, 1 µg per mouse of IL-7 (R&D) was injected intraperitoneally daily from day $-1$ to 5 after transplantation. All flow cytometric analysis were performed with LSR II cytometer (BD Biosciences).

**Treg cells in vitro activation.** In the experiments where Treg cells were adoptively transferred, purified Treg cells were cultured for 2–4 days in RPMI with 10% fetal bovine serum and IL-2 and stimulated with anti-CD3/CD28 beads (Thermo-Fisher). At the end of culture Treg cells were analysed for purity (FoxP3 > 95%).

**Immunostaining.** Frozen BM sections were prepared and immunostained according to the Kawamoto Method[41]. Thin sections were fixed with 4% paraformaldehyde or ethanol. For IL-7 staining, anti-IL-7 antibody (Abcam) and cell permeablization buffer (eBioscience) were used. Immunofluorescence data were obtained using a TCS SP2 confocal microscope (Leica) and analysed by image J (1.47v, National Institutes of Health).

**Multiplex cytokine assays.** Serum was collected from mice at the indicated time points after DT treatment or transplantation. Twenty different cytokines were analysed in a multiplex assay system (Cytokine Mouse 20-plex Panel for the Luminex platform, LMC0006, Invitrogen) and quantified using the Luminex 200 system (Luminex).

**qRT-PCR.** BM derived stromal cells from FTR mice with or without DT treatment were sorted by FACS and mRNA was purified using RNAeasy mini kit (Qiagen). cDNA was generated with the SuperScript III First-Strand Synthesis System for RT-PCR (Invitrogen) according to the manufacturer's protocol. Real time quantitative PCR was performed with specific Taqman probe (Applied Biosystems, catalogue number; #4331182). Mm99999915_g1 for gapdh, Mm00445553_m1 for cxcl12, Mm00442972_m1 for kitl, Mm01295803_m1 for il-7 were used. All primer sequences are proprietary. ABI Prism Fast PCR system (Applied Biosystems) was used for detection.

**Statistical analysis.** In survival experiments log-rank test (Kaplan Meyer analysis) was used. In chimerism analysis two-way ANOVA with bonferroni post-test was used, while for all the other comparisons the two-tail Student's t-test was used. Error bars in all graphs represent mean and standard deviation. All statistical analysis were performed with Prism 6 (GraphPad Software).

**Data availability.** All the relevant data in this study within the article and its Supplementary Files are available from the authors upon request.

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

## Acknowledgements

We thank the Veterinary Service Center at Stanford University and the Stanford Shared FACS Facility. This research was supported by P01 HL075462 from the National Heart, Lung and Blood Institute. A.P. from Associazione Italiana per la Ricerca sul Cancro—AIRC and American Society for Blood and Marrow Transplantation—ASBMT, H.N. received funding from Daiichi Sankyo Foundation of Life Science, and A.B. from the Interdisciplinary Center for Clinical Research (IZKF) Würzburg, Germany.

## Author contributions

A.P. and H.N. designed, performed all experiments and wrote the manuscript. J.B. and Y.C. performed in vitro analysis. H.S.K. and K.T. designed and performed stromal cell purification. J.W. and J.S. analysed and interpreted data. R.S.N. interpreted the data, provided overall research supervision and wrote the manuscript.

## Additional information

**Competing interests:** The authors declare no competing financial interests.

