## [Peer Review File · Nature Communications]

Reviewers' comments:

Reviewer #1

Expert in B cell lymphopoiesis

(Remarks to the Author):

In their paper, Pierini provide evidence that regulatory T cells are necessary for maintaining B lymphopoiesis. In the absence of T regs, there was a striking and specific loss of early B cell progenitors. Add back experiments with T regs also strongly suggested a specific role for T regs in protecting early B cell progenitors. Defective B lymphopoiesis was correlated with diminished IL-7 expression in the BM by ICAM1+ perivascular cells. Furthermore, administration of IL-7 partially restored B cell development in T reg depleted mice. The observations are novel and unexpected. In general, the experimental approach was very solid and conclusions supported by the results. However, there are a few issues that need to be resolved.

1. There needs to be a DT alone control to ensure B cell depletion is not due to non-specific toxicity. This would complement the experiments in Figure 1. Also, there needs to be details in the methods about how mice were treated with DT.

2. Some results appear inconsistent. For example, all lineages were diminished in the HSC reconstitutions. This did not match the straight in vivo phenotype nor did it match the results obtained with TBM transplantations. Likewise, in some cases of allogeneic transplantation there was a decrease in multiple lineages (Figure 4E). These differences in outcome need to be discussed.

3. Some experiments are difficult to follow. For example in Figure 5, T cell depleted BM was transferred into a *rag2^{-/-}gc^{-/-}* host. However, CD4 and CD8 T cells were readily detected. Was the BM really T cell depleted? Is there some evidence that T regs could be mediating their effect through cells other than CD4 effectors such as NK cells?

4. There are no experiments demonstrating that the effect of Tregs are dependent, or not, on T effector cells. Such an experiment would strengthen the mechanistic aspect of the paper.

Reviewer #2

Expert in Tregs

(Remarks to the Author):

Treg cells have been demonstrated to be important in the control of B-cell antibody production in the periphery. However, their role in the maintenance of immune homeostasis in the bone marrow is not well understood. The manuscript by Pierini et al. demonstrates that Treg depletion cause defects in the generation of B-cells and particularly focus on this in the context of B-cell reconstitution after transplantation. They show that this is due to loss of IL-7 production by ICAM1+ perivascular stromal cells, although the exact link between loss of Treg function and loss of function of these cells is unclear.

Overall the paper is well written and the data well-presented and broadly convincing. The subject is of interest and the results have some clear clinical implications and as such I have no major concerns.

Comments:

1: Particularly in the discussion section some of the references are linked to the wrong statements. line 415: "Treg can regulate B cell function through direct B cell killing(Nguyen et al.,2007; Pierini et al., 2015) or via CTLA4-mediated interactions(Nguyen et al.,2007; Pierini et al., 2015) and by limiting B cell production of autoantibodies(Ludwig-Portugall et al., 2008; Zhao et al., 2006)."

The correct papers referring to direct B-cell killing are the Ludwig-portugall and Zhao et al. while the

CTLA-4 papers are the Sage et al. and Wing et al. papers currently on line 423. The references are correct but in the wrong place. This happens at a few points so the authors should double check the placement of all references.

2: Figure S7. The IL7 panel appears to be completely blank. I can see some red coloration on the merged image so assume this may be a problem with the file rather than the data itself. Please check this.

3: A few typographical errors are present, e.g. "." line 362, "IL-7 in injected into" line 371.

Reviewer #3

Expert in haematopoiesis and HSCs

(Remarks to the Author):

The authors present in this paper an interesting concept suggesting that Foxp3+ T reg promote B cell differentiation by regulating niche cells in the bone marrow. The conclusions are generally well supported by the data. However, there are some important concerns that need to be addressed before publication.

1- The title of the article is overstating the paper content "Foxp3+ regulatory T cells maintain the bone marrow microenvironment...", the authors only show that it affects the expression of IL7 from stromal ICAM1+ cells.

2- In several depletion experiments, the authors fail to include a DT control (wild-type mice DT treated).

3- It is unclear if the phenotypes observed (HSC loss of quiescence and lower reconstitution of B cells) after Foxp3+ cells depletion are also a consequence of the increase in the concentration of inflammatory cytokines due to T cell activation.

4- It is difficult to reconcile the results from Fig 2 B and Fig S1G, since both donor cells were collected from Foxp3 depleted mice and the only difference between competitive experiments seems to be that in Fig S1G HSCs were sorted and in Fig 2B is total BM cells.

5- Does Foxp3+ cells depletion affect the total BM cellularity? This should be included in Fig 1.

6- Consistency in the transplantation experiments is needed in the time points of analysis; the standard for testing functional HSC activity should be 16 weeks, minimum. The data in the article range between 6-18 wks.

As a minor related point, the engraftment graphs shouldn't include time point 0 since the chimerism evaluation started only at week 2.

7- In Figure S2 B(i) and (ii) the absolute Pro-B cell number graphs are exactly the same, this seems a duplication of figures.

8- In Figure S7 A there is no IL7 staining, just a black square, but in the overlay image IL7 is present?

9- Sentence #228-229, regarding experiment FigS4; if the goal of the experiment is to use purified allogeneic HSCs, the authors shouldn't sort LSK, these are not the same as purified HSCs.

10- In the model (Fig7), the authors have activated T cells acting directly in ICAM1+ stromal cells to modulate IL-7 expression, this is speculative and the authors don't have data supporting this, perhaps should add question marks in the drawing. In addition, the authors should replace the sinus legend by blood vessel, since they don't show that the distribution of ICAM1+ cells is exclusive of sinusoids or other vessel types like arterioles

REVIEWERS' COMMENTS:

Reviewer #1 (Remarks to the Author):

With revision, the data supports the model as presented in Figure 7. I have no remaining reservations regarding the paper.

Reviewer #2 (Remarks to the Author):

I thank the authors for their reply and am happy with the manuscript at this stage.

Reviewer #3 (Remarks to the Author):

The authors have addressed the issues.

Reviewers' comments:

Reviewer #1

Expert in B cell lymphopoiesis

(Remarks to the Author):

In their paper, Pierini provide evidence that regulatory T cells are necessary for maintaining B lymphopoiesis. In the absence of T regs, there was a striking and specific loss of early B cell progenitors. Add back experiments with T regs also strongly suggested a specific role for T regs in protecting early B cell progenitors. Defective B lymphopoiesis was correlated with diminished IL-7 expression in the BM by ICAM1+ perivascular cells. Furthermore, administration of IL-7 partially restored B cell development in Treg depleted mice. The observations are novel and unexpected. In general, the experimental approach was very solid and conclusions supported by the results. However, there are a few issues that need to be resolved.

1. There needs to be a DT alone control to ensure B cell depletion is not due to non-specific toxicity. This would complement the experiments in Figure 1. Also, there needs to be details in the methods about how mice were treated with DT.

We included the experimental schemes in Figures where required (Figure 1A, Figure 2A, Figure 3A, Figure 4A, Figure 5A) in order to clearly show the detailed protocol used. We also added the WT and WT +DT groups in figures 2 and 3 showing that the phenomenon was not due to non specific toxicity of DT.

2. Some results appear inconsistent. For example, all lineages were diminished in the HSC reconstitutions. This did not match the straight in vivo phenotype nor did it match the results obtained with TBM transplantations. Likewise, in some cases of allogeneic transplantation there was a decrease in multiple lineages (Figure 4E). These differences in outcome need to be discussed.

All data consistently suggested that the expanded CD34-LSK population (phenotypic LT-HSC) from Treg depleted mice was not functional although the total number of functional HSC was unchanged. As shown in Figure S1, although "phenotypic" LT-HSCs were expanded, the data showed that such population had decreased reconstitution capacity in a per cell basis after Treg depletion. On the other

hand, the results from the competitive repopulation analysis using whole bone marrow cells clearly showed that the “functional” HSC number is not significantly changed. Because there are no differences in B cell differentiation capacity of functional HSC in the bone marrow after Treg depletion, we concluded that the bone marrow environment for B cell differentiation was affected after Treg depletion.

In the transplant using FTR mice as recipients (Fig 3, 4, and Figure S4), we consistently observed decreased donor chimerism in lymphoid lineages, especially in B cell lineages. As you mentioned, we observed engraftment failure or decreased chimerism in multiple lineages (especially in lymphoid lineages) in the allogeneic transplant model (Figure 4, Figure S4), which might be due to stronger immune reactions than those in the transient Treg depletion model (Figure 1) or syngeneic transplant model (Figure 3). We added a sentence in the text (Lines 232-233) and we mentioned these issues in the discussion (Lines 389-393).

3. Some experiments are difficult to follow. For example in Figure 5, T cell depleted BM was transferred into a *rag2^{-/-}gc^{-/-}* host. However, CD4 and CD8 T cells were readily detected. Was the BM really T cell depleted? Is there some evidence that Treg could be mediating their effect through cells other than CD4 effectors such as NK cells?

In Figure 5, we used T cell-depleted BM cells isolated by negative selection of CD4/CD8 beads. We confirmed T cell depletion before transplantation. T cell contamination in TCD BM was <0.5% of total cells and we reported it in the main manuscript (Lines 542-543). We also added a sentence in the text (Lines 263-264).

While Treg have been shown to interact and regulate function of different immune cells, it is reasonable to believe that during the engraftment phase after transplantation Treg can control host versus graft reactions that are mainly sustained by residual host T cells. Recently some authors reported a key role for subsets of host NK cells in rejection (Sun, Blood 2012), therefore, it is impossible to exclude that Treg could also limit their function. We mentioned these issues in Discussion (Line 454-458).

4. There are no experiments demonstrating that the effect of Tregs are dependent, or not, on T effector cells. Such an experiment would strengthen the mechanistic aspect of the paper.

In Figure S1, we showed the activation status of residual BM T cells after Treg depletion. According to the several immune phenotypic changes such as CD44, CD62L, and CD69, we concluded the host T

cells were activated (Page 7). We agree that it would be possible that Treg react with the hematopoietic stem and progenitor cells directly, independent on effector T cells. Further, considering the results that hematopoietic reconstitution after transplant was better in immune deficient mice (Balb/c Rag2gamma-chain-/- recipients), Treg might play some role in engraftment, independent on the regulation of residual host derived immune system. We speculated that the activation of donor derived T cells was regulated by the adoptive Treg transfer in this model (Figure 5D). We mentioned these issues in the Discussion (Lines 468-469). Thus, Treg appear to exert their role in the BM through different mechanisms that cannot be entirely described in our manuscript because of their complexity. As we could clearly observe that host T cells change their activation status in the presence or not of Treg, we believe that our proposed model is in any case well supported by the data (Figure 7).

Reviewer #2

Expert in Tregs

(Remarks to the Author):

Treg cells have been demonstrated to be important in the control of B-cell antibody production in the periphery. However, their role in the maintenance of immune homeostasis in the bone marrow is not well understood. The manuscript by Pierini et al. demonstrates that Treg depletion cause defects in the generation of B-cells and particularly focus on this in the context of B-cell reconstitution after transplantation. They show that this is due to loss of IL-7 production by ICAM1+ perivascular stromal cells, although the exact link between loss of Treg function and loss of function of these cells is unclear.

Overall the paper is well written and the data well-presented and broadly convincing. The subject is of interest and the results have some clear clinical implications and as such I have no major concerns.

Comments:

1: Particularly in the discussion section some of the references are linked to the wrong statements. line 415: "Treg can regulate B cell function through direct B cell killing(Nguyen et al.,2007; Pierini et al., 2015) or via CTLA4-mediated interactions(Nguyen et al.,2007; Pierini et al., 2015) and by limiting B cell production of autoantibodies(Ludwig-Portugall et al., 2008; Zhao et al., 2006)."

The correct papers referring to direct B-cell killing are the Ludwig-Portugall and Zhao et al. while the CTLA-4 papers are the Sage et al. and Wing et al. papers currently on line 423. The references are

correct but in the wrong place. This happens at a few points so the authors should double check the placement of all references.

2: Figure S7. The IL7 panel appears to be completely blank. I can see some red coloration on the merged image so assume this may be a problem with the file rather than the data itself. Please check this.

3: A few typographical errors are present, e.g. ". ." line 362, "IL-7 in injected into" line 371.

We apologized for these errors in references and image data. We corrected these errors in the revised MS and corrected the image in Figure S7.

Reviewer #3

Expert in haematopoiesis and HSCs

(Remarks to the Author):

The authors present in this paper an interesting concept suggesting that Foxp3+ Treg promote B cell differentiation by regulating niche cells in the bone marrow. The conclusions are generally well supported by the data. However, there are some important concerns that need to be addressed before publication.

1- The title of the article is overstating the paper content "Foxp3+ regulatory T cells maintain the bone marrow microenvironment...", the authors only show that it affects the expression of IL7 from stromal ICAM1+ cells.

We understand the reviewer's concern regarding the title; thus we specified in the discussion that our study describes only one possible mechanism of Treg function in the bone marrow (Line 454-458). At the same time our data show that Treg maintain bone marrow microenvironment by controlling cytokine production thus promoting B cell differentiation and reconstitution after transplant, therefore we believe that the title is enough representative

2- In several depletion experiments, the authors fail to include a DT control (wild-type mice DT treated).

We routinely performed a DT control group in our experiments but deleted this group in the publication for ease of presentation. However, due to your and Reviewer #1's suggestions we added the WT and WT+DT groups in figures 2 and 3 showing that the phenomenon was not due to non specific toxicity of DT (Lines 115-117).

3- It is unclear if the phenotypes observed (HSC loss of quiescence and lower reconstitution of B cells) after Foxp3+ cell depletion are also a consequence of the increase in the concentration of inflammatory cytokines due to T cell activation.

Our data demonstrated that after Treg depletion: 1) "Phenotypic" LT-HSC (CD34-LSK) are expanded and they lost the quiescence and reconstitution capacity (Fig1 and FigS1); 2) The number of "functional" HSC was not significantly changed, considering the comparable chimerism in TBM transplant (Fig2); 3) Levels of several inflammatory cytokines including IFN γ and TNF α are increased (FigS1); 4) The residual T cells are activated (Fig S1).

In the previous literature, excess of IFN or TNF signaling affects the function of HSC and induces their exhaustion (Sato et al. Nature Med. 2009, Pronk et al. J Exp Med. 2011). Their data imply that one of the major factors in our Treg depleted model might be IFN and TNF from activated T cells. However, as shown in Figure 2, the number of functional HSC was not changed in Treg depleted mice and a defect of B cell differentiation was instead observed. Moreover we added data about cytokine concentration in serum of transplanted mice in syngeneic conditions in Figure S2 and described them in the text (Lines 196-198). Taken together, these data suggest that the phenotype in Treg depleted mice cannot be explained only by the direct activation of HSC/HPC by inflammatory cytokines. Therefore we can hypothesize that the functional defect in bone marrow environment, especially in IL-7 secreting ICAM1+ stromal cells, could be due to "self" or "allo"-reactive activated T cells.

We agree that the Treg depleted model is not an appropriate model for investigating the effect of specific inflammatory cytokines in the hematopoietic system (Fig S1). However, these points were beyond the scope of this manuscript. We focused on the role of Treg in hematopoiesis after transplant,

rather than the effect of a specific inflammatory cytokines. We added to the discussion about these issues (Lines 378-393).

4- It is difficult to reconcile the results from Fig2 B and Fig S1G, since both donor cells were collected from Foxp3 depleted mice and the only difference between competitive experiments seems to be that in Fig S1G HSCs were sorted and in Fig 2B is total BM cells.

As discussed above, all of the data consistently suggested that some expanded CD34-LSK population (phenotypic LT-HSC) from Treg depleted mice were not functional although total number of functional HSC was unchanged.

As shown in figure S1, although “phenotypic” LT-HSCs were expanded, the data showed that such population had decreased reconstitution capacity in cell per basis after Treg depletion. On the other hand, the results from the competitive repopulation analysis using whole bone marrow cells clearly showed that the “functional” HSC number is not significantly changed. Because there were no differences in B cell differentiation capacity of functional HSC in the bone marrow after Treg depletion, we concluded that the bone marrow environment for B cell differentiation was affected after Treg depletion. We corrected the text in order to better clarify the concept (Lines 168-171).

5- Does Foxp3+ cells depletion affect the total BM cellularity? This should be included in Fig 1.

We added new data regarding cellularity in Treg depleted mice (Fig 1, Lines 124-125).

6- Consistency in the transplantation experiments is needed in the time points of analysis; the standard for testing functional HSC activity should be 16 weeks, minimum. The data in the article range between 6-18 wks.

As a minor related point, the engraftment graphs shouldn't include time point 0 since the chimerism evaluation started only at week 2.

We added the data at 16 weeks in Figure 2. We adjusted consistency in transplantation experiments with a maximum follow up of 12 weeks. We understand that 16 weeks is a good time point for assessing functional HSC activity, but in all our transplantation experiments (with the exception of Figure S4) we transferred donor BM or TCD BM and we observed main differences in B cell reconstitution in the early phase after transplantation. Thus, we believe that 12 weeks are enough to observe a delayed B cell reconstitution in the absence of Treg. We also excluded time point 0 as suggested.

7- In Figure S2 B(i) and (ii) the absolute Pro-B cell number graphs are exactly the same, this seems a duplication of figures.

8- In Figure S7 A there is no IL7 staining, just a black square, but in the overlay image IL7 is present?

We apologized for these errors. We corrected Figure S2B and Figure S7

9- Sentence #228-229, regarding experiment FigS4; if the goal of the experiment is to use purified allogeneic HSCs, the authors shouldn't sort LSK, these are not the same as purified HSCs.

We agree that LSK cells include hematopoietic progenitor cells, which lack the reconstitution capacity. The aim of this experiment is to confirm that Treg affect the precise differentiation into B cell lineages from multipotent progenitors including HSC. Because mature donor cells including mature B cells and T cells were not included in the donor graft, these data suggested that Treg depletion can modify the BM environment for B cell differentiation from HSC/HPC. We described LSK cells as HSC/HPC (hematopoietic progenitor cells, Line 243).

10- In the model (Fig7), the authors have activated T cells acting directly in ICAM1+ stromal cells to modulate IL-7 expression, this is speculative and the authors don't have data supporting this, perhaps should add question marks in the drawing. In addition, the authors should replace the sinus legend by blood vessel, since they don't show that the distribution of ICAM1+ cells is exclusive of sinusoids or other vessel types like arterioles

We agree that the model we showed in Figure 7 was too speculative. We newly put the question mark and changed “Blood vessels” from “sinus”, as suggested.

We thank the editor and the reviewers for the positive revision of our manuscript.

Reviewers' comments:

Reviewer #1

Expert in B cell lymphopoiesis

(Remarks to the Author):

With revision, the data supports the model as presented in Figure 7. I have no remaining reservations regarding the paper.

We thank the reviewer for approving our revision

Reviewer #2

Expert in Tregs

(Remarks to the Author):

I thank the authors for their reply and am happy with the manuscript at this stage.

We thank the reviewer for approving our revision

Reviewer #3

Expert in haematopoiesis and HSCs

(Remarks to the Author):

The authors have addressed the issues.

We thank the reviewer for approving our revision